# OpenReview forum: "Hijacking JARVIS: Benchmarking Mobile GUI Agents against Unprivileged Third Parties"
_ICLR.cc/2026/Conference — ICLR 2026 Conference Withdrawn Submission_

### Official Review · Reviewer_UHX2 · 2025-10-29

**Soundness:** 3
**Presentation:** 3
**Contribution:** 2
**Rating:** 4
**Confidence:** 3

**Summary:**

The paper studies robustness of mobile GUI agents against misleading content originating from unprivileged third parties and introduces AgentHazard, a configurable attack-simulation framework that patches adversarial content into Android apps at run time. Built on this, the authors assemble a benchmark with a dynamic interactive environment and a static dataset containing benign and adversarial variants. Two metrics are used: Success Rate Drop (∆SR) and Misleading Rate (MR), where MR counts whether an agent executes a predefined misled action tied to injected content. Experiments span six representative mobile agents (e.g., M3A, T3A, AutoDroid, AriaUI, UI-TARS-1.5) and five LLM backbones across text/vision/multimodal settings. Results show substantial vulnerability (average MR ≈ 42.1% dynamic; 40.7% static), with multi-modal inputs generally more susceptible, UI-TARS-1.5 relatively robust, and Claude-4-sonnet outperforming GPT-4o variants on the static dataset. A small adversarial training study with Qwen2.5-VL suggests modest robustness gains but no fundamental fix.

**Strengths:**

Clear problem framing around unprivileged third-party content and the resulting need to test agents outside “clean” environments; Figure 1 offers a compelling case of destructive behavior induced by short misleading strings.

AgentHazard is described as a practical, scalable simulator that modifies both screenshots and the UI element tree in real time via an Android Accessibility-based tool and a cooperating Python attack module; the configuration language (Appendix D, Listing 1) is concrete and appears easy to author.

Useful findings for the community: multi-modal inputs increase susceptibility on average; robustness depends strongly on both model family and GUI-specific training; simple popup-style attacks are much easier to detect than natively rendered modifications (Table 5).

**Weaknesses:**

Although the paper claims feasibility for unprivileged third parties, the simulator “patches adversarial content both on the screen and the structured UI element tree in real time.” Real third parties can plausibly control in-app content regions (e.g., posts, product titles), but they cannot modify another app’s UI tree; the paper should more clearly separate (i) realistic adversary control over pixels/strings inside third-party-controlled regions from (ii) the tool’s additional modifications to the UI tree used to keep text-based perception consistent. As written, the phrase “as native GUI elements” risks overstating real-world feasibility.

Step 4 (Attack Rule Generation) relies on LLMs to produce adversarial strings given prompts P, but the exact model, decoding parameters, and filtering are not provided in the main text; inter-annotator criteria for “feasibility” (Step 2) and quality control for rules Rsuccess/Rattack are also not quantified (e.g., agreement rates).

Stealthiness study (Table 5) depends on a single LLM-based detector and reports a relatively high 10.3% “No Attack” detection rate without confidence intervals or sample counts; the detector may not be a reliable surrogate.

In addition, I have some concerns on the novelty of this paper, I acknowledge it may be interested by someone working on the GUI agents (the practical contribution is more than the technical contribution).

**Questions:**

You state that “content modifications are designed to be feasible for unprivileged third parties” while AgentHazard also modifies the UI element tree. In real apps, third parties cannot alter another app’s UI tree. Please clarify precisely which parts of your injected content are intended to model realistic third-party control and which are simulator conveniences to keep text-only perception in sync. Can you report results where only on-screen pixels are modified but the UI tree is left untouched?

The paper positions itself as a first systematic investigation for mobile agents. Could you delineate, in the main text, how your dynamic and static components differ from prior mobile-agent robustness evaluations you cite (e.g., MobileSafetyBench) to avoid ambiguity in the novelty claim?

---

### Official Review · Reviewer_F9Tf · 2025-11-01

**Soundness:** 2
**Presentation:** 3
**Contribution:** 2
**Rating:** 2
**Confidence:** 3

**Summary:**

This paper investigates the vulnerability of mobile GUI agents to misleading content from untrusted third-party sources. The authors develop AgentHazard, a framework for injecting adversarial content into Android applications without requiring elevated privileges, and construct a benchmark comprising 122 dynamic tasks and over 3,000 static scenarios. Through evaluation of six mobile GUI agents and five backbone LLMs, they demonstrate that current agents exhibit significant vulnerability, with average misleading rates exceeding 40% in both dynamic and static settings. While the paper addresses a practically important problem and provides a useful evaluation framework, it suffers from limited scope in attack modeling, insufficient depth in analyzing root causes, and superficial treatment of defense mechanisms.

**Strengths:**

1. The threat model is realistic and well-motivated, focusing on unprivileged attackers controlling legitimate third-party content rather than privileged access.
2. The dual evaluation, dynamic interaction and static state-rule analysis, offers complementary insights. Broad testing across six agents and five LLMs, including multimodal and text-only models, enhances generalizability.
3. Several empirical findings are valuable: visual modalities show higher vulnerability than text, different LLMs exhibit varying sensitivity to attack types.

**Weaknesses:**

1. The study only tests two attack types, wrong clicks and early exits, which miss many real threats like phishing, permission abuse, or data leaks. Saying it’s “hard to measure” complex attacks isn’t a good excuse. The focus on single misleading items makes the setup too clean and unrealistic.
2. The paper shows what fails but not why. It never explores why visual models are weaker, maybe attention or data bias? Without tools like attention maps or gradient analysis, the study stays at a surface level.
3. The defense experiment is poorly designed and reuses data from failed samples. It also ignores better-known defenses like careful prompting, confidence checks, or human review.
4. Using GPT-4o to both make and judge attacks causes bias. It may just favor its own style, not real-world tricks.

**Questions:**

See weakness

**Details Of Ethics Concerns:**

An identically titled paper has already been published in the EdgeFM '25 Proceedings. Here is the official publication record: https://doi.org/10.1145/3737902.3768354

---

### Official Review · Reviewer_Nvyy · 2025-11-01

**Soundness:** 3
**Presentation:** 3
**Contribution:** 3
**Rating:** 4
**Confidence:** 2

**Summary:**

This paper provides an investigation into the vulnerability of mobile GUI agents. The main contribution is a benchmark suite including both dynamic emulators and static screenshots. The benchmark evaluates how much GUI agents are affected by content deliberately crafted to deceive them by untrustworthy third-party sources. Test results reveal that all widely used mobile GUI agents are vulnerable to misleading third-party content, indicating that their robustness needs improvement.

**Strengths:**

- This paper highlights the importance of the ability of defense against third-party adversarial attacks for GUI agents. And it presents a fine-grained benchmark suite to evaluate this.
- This paper tests several current GUI agents on the benchmark. The significant performance drop highlights the safety problem of current agents.
- This paper provides several findings by studying the impact of multimodal, LLM-backbone, and GUI-specific training on the results.

**Weaknesses:**

- The Evaluation is limited. The paper reveals that the GUI-specific training has an impact on the success rate and the misleading rate. However, only GUI-TARS is tested, which affects the credibility of the results. End-to-end trained models like Mobile-Agent-v3[1], UI-Genie[2] may have different results on the benchmark.

-  The authors argue that, unlike existing attacks on web-based agents, this work focuses on mobile GUI agents, which have higher security requirements and stricter control over user privacy, making existing web-based attack strategies difficult to execute. However, apart from introducing an additional Android application, the proposed method still relies on modifying the screen and XML, and its specific differences from existing attacks are not clear.

Note: I will raise my score if my concerns are addressed.

[1] Ye, Jiabo, et al. "Mobile-agent-v3: Fundamental agents for GUI automation."

[2] Xiao, Han, et al. "UI-Genie: A Self-Improving Approach for Iteratively Boosting MLLM-based Mobile GUI Agents."

**Questions:**

From Table 1, we can observe that the SR of UI-TARS only drops a little with misleading information. However, the Qwen-2.5-VL-7B-Instruct trained with benight data demonstrates a significant performance drop in Table 3. What causes this misalignment? What is the effect of the adversarial training on models that are specially fine-tuned on GUI Agent tasks?

---

### Official Review · Reviewer_4CEc · 2025-11-02

**Soundness:** 2
**Presentation:** 3
**Contribution:** 2
**Rating:** 4
**Confidence:** 4

**Summary:**

This paper presents Hijacking JARVIS, a systematic benchmark for evaluating the robustness of mobile GUI agents against misleading content from unprivileged third parties. The paper proposes AgentHazard, a scalable Android attack simulation framework capable of injecting adversarial text and UI elements without system privileges, and constructs both a dynamic environment and a static dataset. Experiments on several representative agents and five backbone LLMs show that some models are vulnerable. However, the UI-TARS-1.5 agent, which is specifically fine-tuned for downstream tasks involving GUI-related operations, exhibits more robust behavior when confronted with misleading information

**Strengths:**

1. The paper is well-organized and clearly written, making it easy to follow.
2. The combination of a dynamic interactive environment and a large-scale static dataset provides both high ecological validity and reproducibility. This will benefit future research on security and robustness in GUI agents.
3. The paper reveals key trends—particularly that multimodal perception increases susceptibility to misleading content, while domain-specific GUI training enhances robustness—providing actionable insights for future model design.

**Weaknesses:**

1. The most critical question for me is: what is the fundamental difference between third-party attacks and traditional pop-up–based attacks? From the standpoint of agent security and defense, third-party attacks can essentially be viewed as a special case of pop-up attacks. Since prior work has already demonstrated that agents are vulnerable to pop-up–based adversarial content, the novelty of verifying vulnerability to third-party attacks appears relatively limited. In practice, when designing agents, we must consider and defend against high-privilege pop-up attacks, which are generally considered a more powerful and comprehensive threat model. In this case,  is it still necessary to investigate defenses against third-party attacks?
2. Only one GUI-specialized model (UI-TARS-1.5) is tested. Including more domain-finetuned agents would strengthen the conclusions and provide richer insights into effective robustness strategies.
3. AgentHazard currently modifies only textual and UI-tree elements; it does not support manipulation of visual assets (e.g., images, icons) or multi-step dynamic attacks, leaving important real-world attack vectors unexplored.

**Questions:**

Disregarding the issue of third-party permissions, what is the fundamental distinction between third-party attacks and pop-up attacks, and which distinct vulnerabilities of the model does this difference validate?

---

### Note · Authors · 2026-01-17

I have read and agree with the venue's withdrawal policy on behalf of myself and my co-authors.